# Application of Lightning Data Assimilation for the 10 October 2018 Case Study over Sardinia

**Rosa Claudia Torcasio** [1], **Stefano Federico** [1,*], **Silvia Puca** [2], **Gianfranco Vulpiani** [2], **Albert Comellas Prat** [3] **and Stefano Dietrich** [1]

1   ISAC-CNR, via del Fosso del Cavaliere 100, 00133 Rome, Italy; rc.torcasio@isac.cnr.it (R.C.T.); s.dietrich@isac.cnr.it (S.D.)

2   Dipartimento Protezione Civile Nazionale Ufficio III—Attività Tecnico Scientifiche per la Previsione e Prevenzione dei Rischi, 00189 Rome, Italy; silvia.puca@protezionecivile.it (S.P.); gianfranco.vulpiani@protezionecivile.it (G.V.)

3   ISAC-CNR, Strada Prov.le Lecce-Monteroni, 73100 Lecce, Italy; a.comellas@isac.cnr.it

*   Correspondence: s.federico@isac.cnr.it

**Abstract:** On 10 October 2018 an intense storm, characterized by heavy rainfall, hit the Sardinia island, reaching a peak of 452 mm of rain measured in 24 h. Among others, two particularly intense phases were registered between 3 and 6 UTC (Universal Coordinated Time), and between 18 and 24 UTC. The forecast of this case study is challenging because the precipitation was heavy and localized. In particular, the meteorological model used in this paper, provides a good prediction only for the second period over the eastern part of the Sardinia island. In this work, we study the impact of lightning data assimilation and horizontal grid resolution on the Very Short-term Forecast (VSF, 3 and 1 h) for this challenging case, using the RAMS@ISAC meteorological model. The comparison between the 3 h VSF control run and the simulations with lightning data assimilation shows the considerable improvement given by lightning data assimilation, especially for the precipitation that occurred in the eastern part of the island. Reducing the VSF range to 1 h, resulted in higher model performance with a good precipitation prediction over eastern and south-central Sardinia. In addition, the comparison between simulated and observed reflectivity shows an important improvement of simulations with lightning data assimilation compared to the control forecast. However, simulations assimilating lightning overestimated the precipitation in the last part of the day. The increasing of the horizontal resolution to 2 km grid spacing reduces the false alarms and improves the model performance.

**Keywords:** lightning data assimilation; very short-term forecast; numerical weather prediction; RAMS@ISAC

## 1. Introduction

Despite the continuous improvement of weather forecast, the correct prediction of heavy precipitation events is still challenging because of the multitude of scales and physical processes involved in these events [1]. Along with other factors, the improvement of numerical weather prediction depends on the improvement of initial conditions prescribed in numerical models [2,3]. In the case of localized and heavy rainfall, initial conditions must be improved at the local scale too.

Lightning can locate deep convection precisely, both in space and time, with few temporal gaps and requires the transfer of small amounts of data [4]. For these characteristics, lightning can be assimilated into meteorological models to force convection also in the Rapid Update Cycle (RUC) operational context. The Lightning Imager (LI) is already flying aboard geostationary GOES-R [5,6] and FY4 satellites [7], while its launch is planned for 2021 onboard Meteosat Third Generation (MTG)

satellites. This instrument will give a wealth of information on convection over the Mediterranean, and gives an added value to NWP (Numerical Weather Prediction) experiments assimilating lightning.

Lightning is not a model state variable and research is done to convert model variables into flashes or to convert lightning into some other observations which can be assimilated into NWP [8,9].

Thus, several data assimilation techniques have been elaborated both for coarse resolution models (>3 km horizontal resolution), which need a convective parameterization scheme for precipitation, and high-resolution numerical weather prediction models (≤3 km horizontal resolution), which allow or explicitly resolve convective precipitation, the latter resolutions being the focus of this paper.

Considering the lightning data assimilation for coarse resolution models, some authors [10–13] determined a relationship between the rain-rate and the flash rate and assimilated the rain-rate through latent-heat nudging.

In [14], Papadopulos et al. used lightning to locate convection and assimilated relative humidity empirical profiles for these positions. The relative humidity profiles for convective events were obtained from severe weather cases well simulated by Skiron/Eta model [15].

In [4], Mansell et al. modified the Kain–Fritsch [16] cumulus convective scheme to force convection when/where flashes are observed. A similar approach was used by Lagouvarodos et al. [17] and Giannaros et al. [18] into WRF (Weather Research and Forecasting Model) showing the positive impact of lightning data assimilation on the precipitation forecast over Greece.

The location and lightning density are well correlated with radar reflectivity [19] and flashes have been assimilated as a supplement of radar reflectivity [20,21]. In other studies flashes were used as a proxy for radar reflectivity and assimilated with different methods [7,8,22] for convection resolving horizontal resolution.

Fierro et al. [23,24], using WRF, proposed a method to assimilate lightning for convection resolving scales adding water vapor to the model as a function of lightning flash rate and graupel mixing ratio. The same technique was applied by Qie et al. [25] also considering snow, ice crystals, and graupel. In [26] Chen et al. combined the methods used in [23,24] and [25] to comprehensively increase both the water vapor content and ice-phase particles. In [27] and [28] Lynn et al. applied Fierro's technique to show the important and positive impact of the method on the rainfall forecast. They also proposed a method to suppress convection when/where the model predicted spurious convection.

Recently, Wang et al. [29] proposed a method conceptually similar to that of Fierro et al. [23,24] where lightning is used as a proxy for graupel concentration. In this method the graupel mixing ratio is retrieved from total lightning and it is distributed in the vertical with empirical profiles derived from model simulations. A horizontal spread method is utilized to consider the existence of graupel in the adjacent regions of the lightning initiation location and latent heat is adjusted to account for the retrieved graupel.

The direct substitution of water vapor or hydrometeors into NWP can be arbitrary and can lead to inconsistencies with surrounding environment. For this reason, several methods to account for the model balance have been proposed to assimilate lightning by 3D-Var [21], Ensemble Kalman Filter [30,31] and time lagged ensemble [32].

Federico et al. [33] implemented the Fierro's technique [23,24] into the RAMS@ISAC (Regional Atmospheric Modelling System at Institute of Atmospheric Sciences and Climate) showing the significant impact of lightning data assimilation on the Very Short Term (VSF, 6 h) rainfall forecast over Italy. In [34] they applied the method at different forecast ranges (from 3 to 24 h) for 20 moderate-severe precipitation events over Italy. They showed that the lightning data assimilation had a positive impact on the rainfall forecast over Italy up to 12 h forecast.

Dynamical processes are another important element in convective systems because they provide the background for cloud microphysical processes, such as condensation, ice particle growth, and generation of supercooled raindrops. In [35], Wang et al. used a method to adjust the dynamic field in convective clouds based on the relationship between the vertical velocities and lightning [36,37].

Recent studies using lightning data from instruments onboard GOES [38] and FY-4 satellites [7] have also shown the usefulness of lightning data to improve the precipitation forecast with different lightning data assimilation techniques.

In this paper we apply a lightning data assimilation method similar to Fierro et al. [23,24] to the precipitation forecast of a challenging case that occurred over Sardinia on 10 October 2018. This is a challenging case study because two heavy precipitation events (360 and 452 mm/24 h, respectively) were recorded in two localized areas of Sardinia island located less than 50 km apart. Small to moderate precipitation was recorded in other areas of the Island. The RAMS@ISAC model was not able to simulate and sustain the convection without lightning data assimilation for most of the simulated VSF. For these characteristics of the precipitation field, the case study is particularly useful to investigate the following points: (a) the performance of a 1 h Rapid Update Cycle compared to the 3 h Rapid Update Cycle, which is the usual configuration of RAMS@ISAC [20]; (b) the impact of increasing the model horizontal resolution. For the latter point we consider the comparison between 3 and 2 km horizontal resolution. The paper is divided as follows: Section 2 describes some characteristics of the storm at the synoptic scale, highlighting the role of the water vapor plume, connected with the tropics, that fed the storm with humid air masses; Section 3 shows the model setting and the method used in this paper to assimilate lighting; Section 4 shows the results; Section 5 summarizes the main conclusions.

It is important to note that several methods for lightning data assimilation have been used in the literature and each method has its own advantages. A comparison among different methods is not available and requires systematic and demanding study. The basic idea behind the methodology used in this paper is that the assimilation of the water vapor profile has a positive impact on the dynamic background of convection which, in turn, determines the evolution of thermodynamic and microphysical processes. The main advantages of the scheme used in this paper compared to other methods are that the implementation is straightforward and that the method is computationally cheap.

## 2. The Case Study

The event occurred on 10 October 2018 and affected mainly the Sardinia island. Figure 1 shows the rainfall over Italy on 10 October. Heavy rainfall was observed in eastern and south-central Sardinia with maximum values of 360.4 mm/24 h in Tertenia and 452.4 mm/24 h in Santa Lucia di Capoterra, respectively. Abundant precipitation was also observed over the western Alps and Liguria. The precipitation in northwestern Italy, however, occurred mainly in the evening of 10 October, as the forerunner of a meteorological system that affected northern and central Italy on the following day. Rainfall over Sardinia, however, started at the beginning of 10 October, as will be shown later.

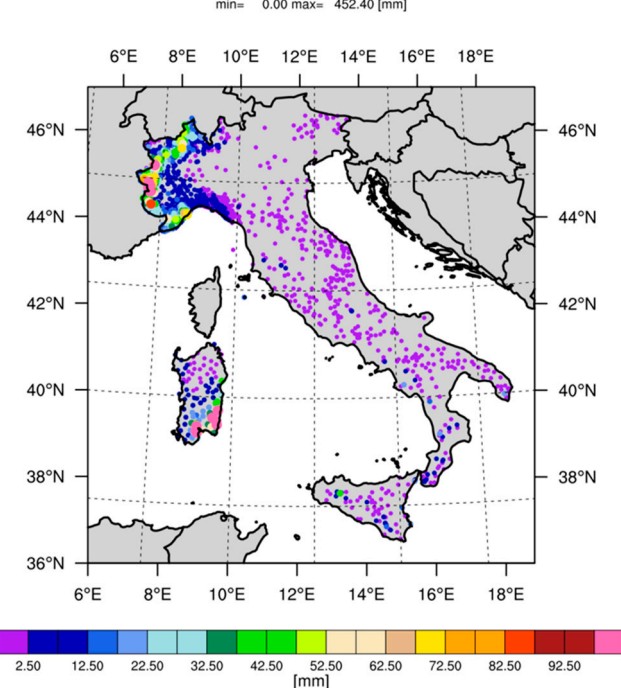

**Figure 1.** Daily precipitation observed on 10 October 2018. The rain gauge network is managed by the Civil Protection Department and has more than 3000 rain gauges over Italy. Only rain gauges recording at least 0.2 mm over the whole day are shown.

Observed rainfall of Figure 1 explains why this case study was selected: two intense and localized precipitation spots were observed over Sardinia, making the forecast of the case study challenging. Furthermore, since flashes are very precise for localizing deep convection, this event is outstanding for exploring the precision of the lightning data assimilation on the rainfall forecast.

From a synoptic point of view (Figure 2, derived from the GFS, Global Forecasting System, operational analyses at the corresponding time), the storm was characterized by a sea level pressure low over the western Mediterranean Sea. Surface winds blew from south to north over Sardinia, advecting warm and unstable air masses over the island, as shown by the high values of equivalent potential temperature south of the region (>330 K). Figure 2a is at 12 UTC on 10 October, nevertheless the synoptic situation did not change during the day.

Figure 2b shows the geopotential, temperature and wind field at 500 hPa and at the same time of Figure 2a. The geopotential height shows three action centers: a minimum over France and two maxima centered over North Africa and Poland. The action of these centers caused a levelled geopotential height over the Tyrrhenian Sea and wind divergence over Sardinia. This divergence helps to sustain convection.

An important meteorological element determining the severity of this case study is the water vapor plume that extends from the tropics towards the Mediterranean Sea (Figure 2c, this figure spans a wider domain compared to Figure 2a,b to better represent the connection of the water vapor plume with the tropics). This plume was located over Sardinia at 12 UTC on 10 October, nevertheless we show it at 00 UTC on this day because its connection with the tropics is more apparent. Thus, convection over Sardinia developed in a humid environment which helped to sustain the convection and determined the severity of this case study.

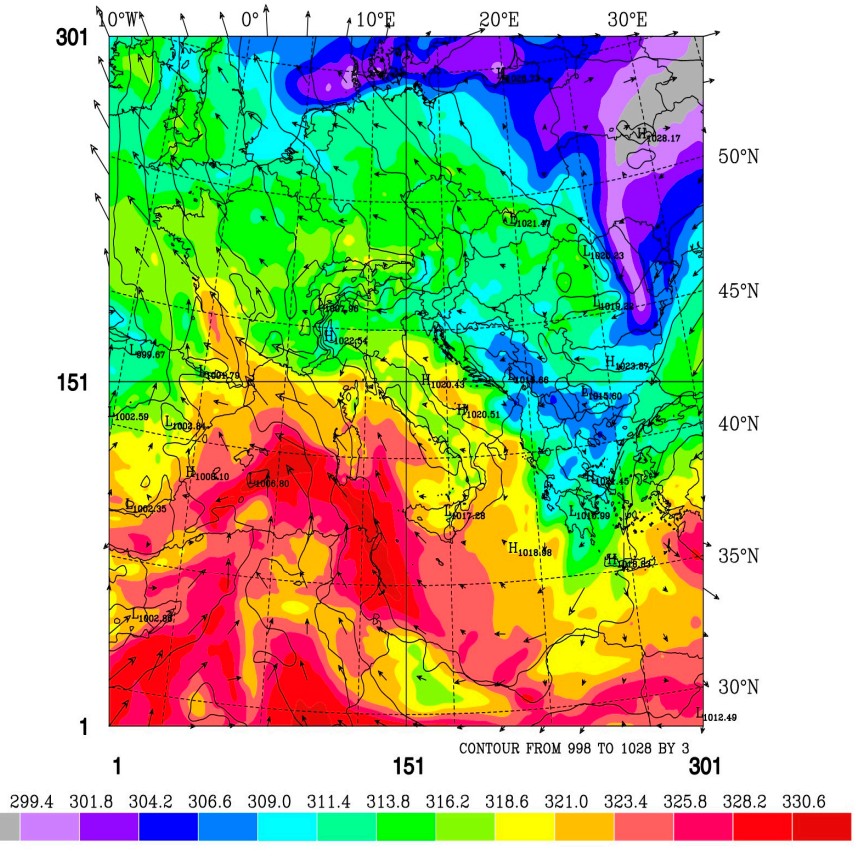

(a)

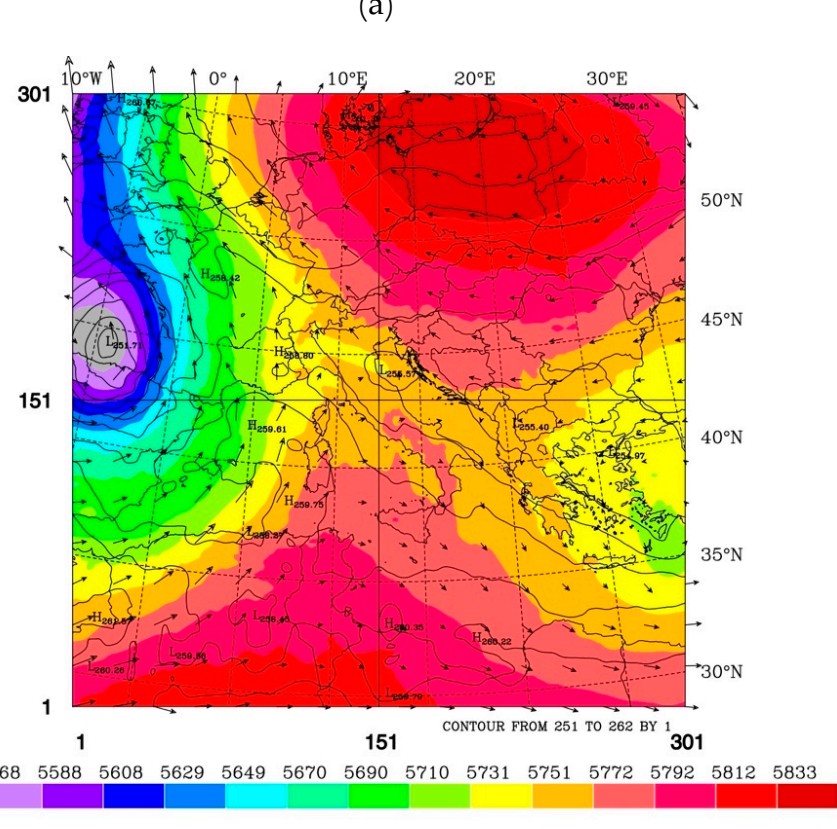

(b)

**Figure 2.** *Cont.*

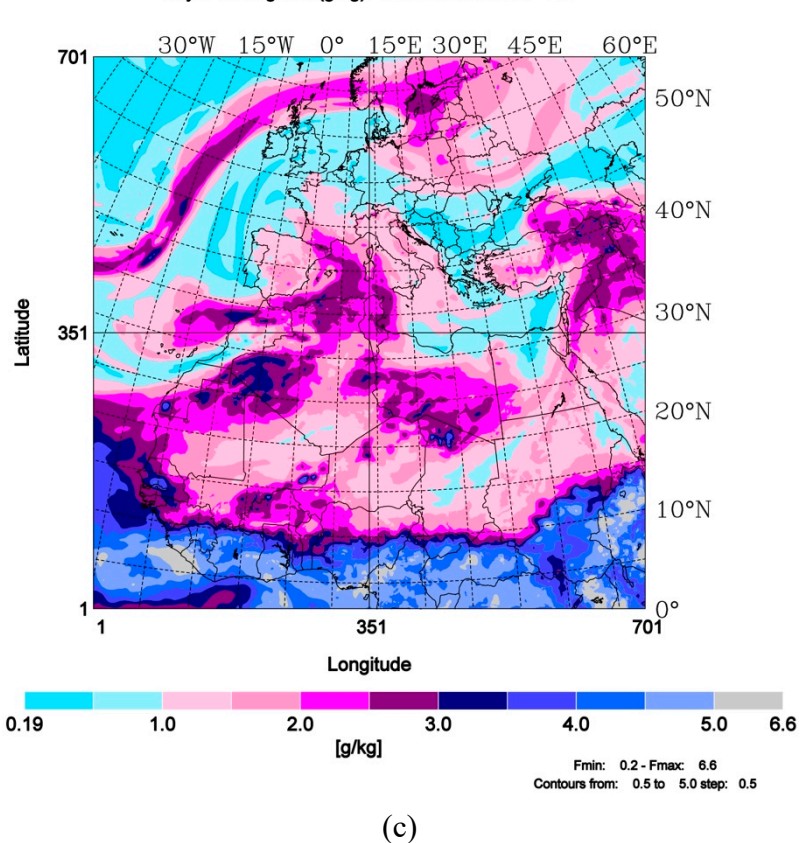

(c)

**Figure 2.** (**a**) Equivalent potential temperature (filled contours, (K)), sea surface pressure (contours, (hPa)), and wind vector (one arrow every 15 grid points in both directions, maximum value 16 m/s) at 12 UTC (Universal Coordinated Time) on 10 October 2018; (**b**) geopotential height (filled contours, (gpm)), temperature (contours, (K)) and wind vectors (one arrow every 15 grid points in both directions, maximum value 28.8 m/s) at 12 UTC on 10 October 2018; (**c**) averaged water vapor mixing ratio between 3 and 10 km height at 00 UTC on 10 October 2018. The labels along the bottom x-axis and the left y-axis show the grid-point number.

In the following sections, we consider two 3 h Very Short-term Forecast (VSF), the first between 03 and 06 UTC on 10 October and the second between 21 UTC on 10 October and 00 UTC on 11 October. The CTRL forecast, which did not assimilate lightning, did not predict the first phase, while predicted the second phase well. Therefore, the comparison of the two phases shows the impact of lightning data assimilation for two different performances of the CTRL forecast.

Figure 3a shows the precipitation for the first phase. Rainfall was intense over eastern and south-central Sardinia. Some rain was observed over western Alps, while precipitation was negligible in the rest of Italy. Figure 3b shows the CAPPI (Constant Altitude Plan Position Indicator) at 04 UTC and 5 km height. Convective cells are apparent over the central part of southern Sardinia and over eastern Sardinia, with values up to 50 dBz.

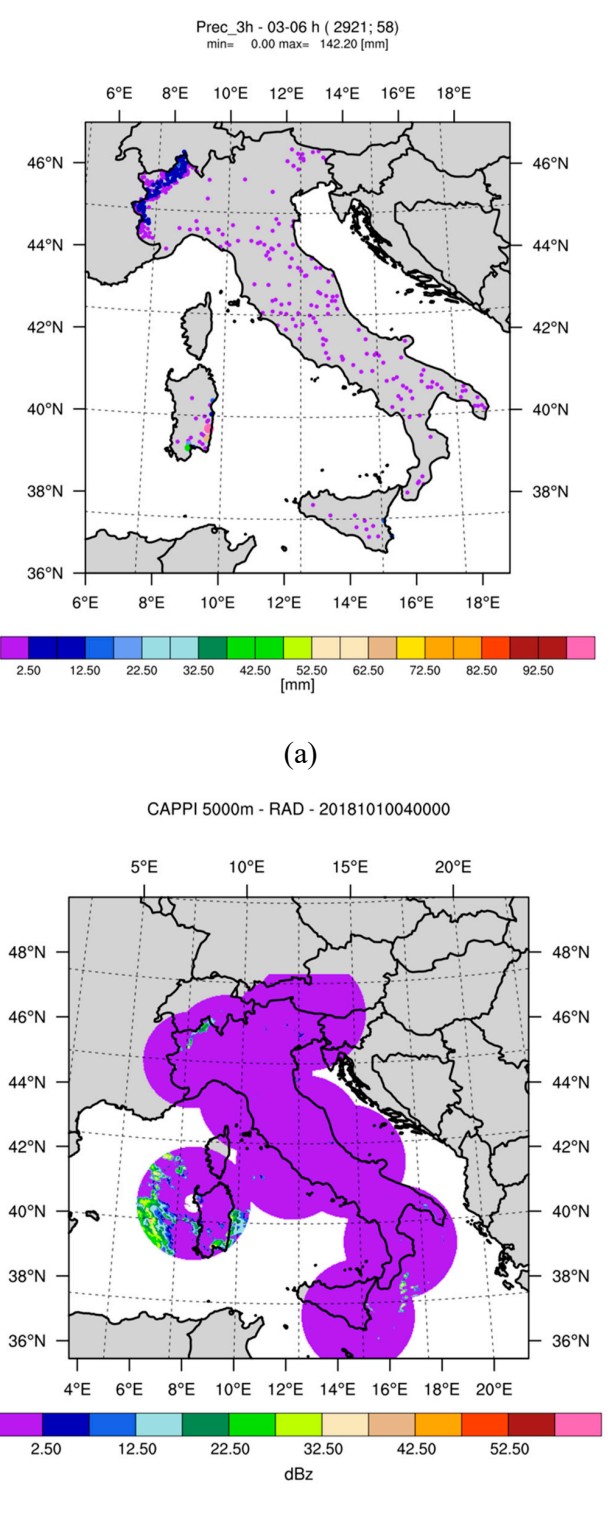

(a)

(b)

**Figure 3.** (**a**) Rainfall observed by the Italian rain gauge network between 03 and 06 UTC on 10 October; (**b**) CAPPI (Constant Altitude Plan Position Indicator) at 04 UTC on 10 October and at 5 km height. Radar composite is obtained from the 22 radars of the Italian network.

Figure 4a shows the observed rainfall between 21 UTC on 10 October and 00 UTC on 11 October. Intense precipitation occurred over Sardinia, Liguria, and western Alps (Piedmont and Valle d'Aosta). The CAPPI at 21 UTC and 5 km height (Figure 4b) shows a wide cloud system extending over southern

and eastern Sardinia and the meteorological system over northwestern Italy, as the forerunner of the meteorological system, which produced heavy precipitation over northern and central Italy on 11 October.

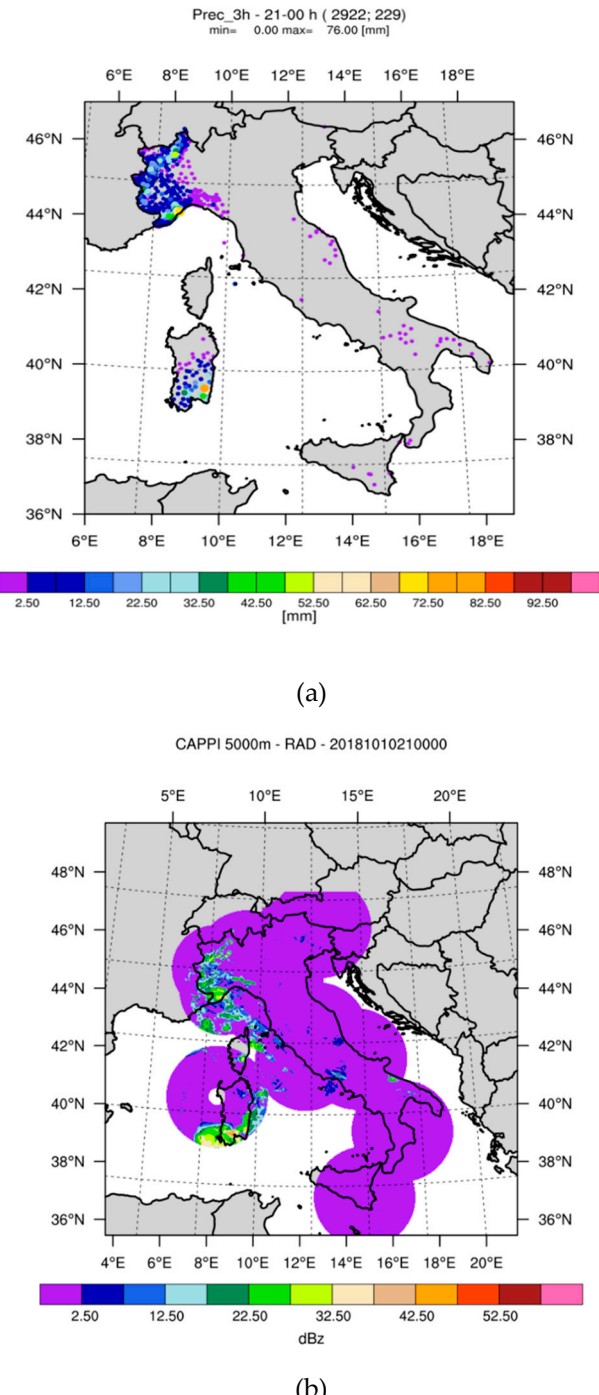

(a)

(b)

**Figure 4.** (**a**) rainfall observed by the Italian rain gauge network between 21 UTC on 10 October and 00 UTC on 11 October; (**b**) CAPPI at 21 UTC on 10 October and at 5 km height. Radar composite is obtained from the 22 radars of the Italian network.

The comparison between Figures 3a and 4a is consistent with the fact that intense precipitation over Sardinia occurred during the whole of 10 October, while intense precipitation over the Alps and Liguria occurred mainly during the evening of 10 October.

## 3. Data and Methods

### 3.1. RAMS@ISAC Model

The RAMS@ISAC model was used in this work. This model is based on the RAMS 6.0 model [39], with a number of modifications and improvements: (a) the WSM6 (WRF single moment six species, [40]) microphysical scheme [41] was added to other microphysics schemes available in RAMS; (b) the model implements a method for predicting lightning [42], following Dahl et al. [43]; (c) lightning data assimilation was implemented [33], following the method of Fierro et al. [23,24]. Recent developments of the RAMS model, not used in this work, are shown in [44].

Two different configurations are used (C1 and C2). Each configuration has two domains, the first domain covers central Europe, while the second domain covers the Italian territory (Figure 5, Table 1). The first domain and grid resolution (10 km, R10) are the same for both configurations.

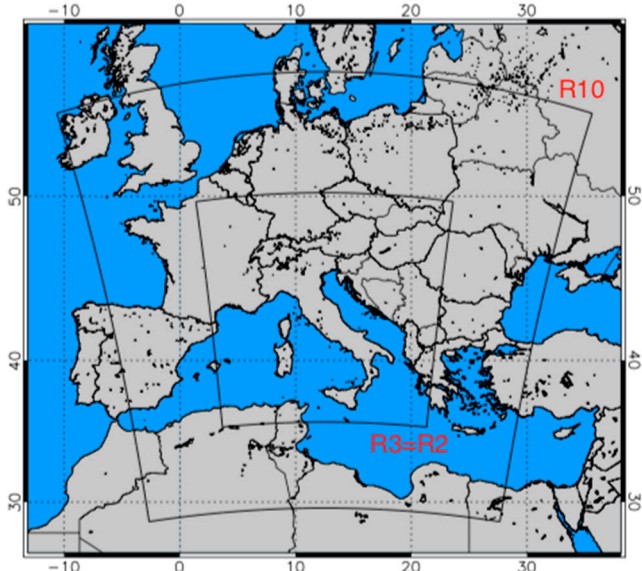

**Figure 5.** Model domains used for the simulations. The extension of R2 and R3 are similar (about 1600 km).

**Table 1.** Basic parameters of the RAMS@ISAC grids R10, R3, and R2, corresponding to the two configurations C1 and C2. C1 uses R10 and R3, C2 uses R10 and R2. NNXP and NNYP represent the number of grid points in the WE and NS directions, respectively. NNZP is the number of vertical levels. DX and DY are the grid spacing sizes in the WE and SN directions. Lx, Ly, and Lz are domain extensions in NS, WE, and vertical directions. CENTLON (longitude of the grid center) and CENTLAT (latitude of the grid center) are the coordinates of the grid centers.

|  | R10 | R3 | R2 |
|---|---|---|---|
| NNXP | 301 | 535 | 803 |
| NNYP | 301 | 535 | 803 |
| NNZP | 42 | 42 | 42 |
| Lx | 3000 km | 1605 km | 1606 km |
| Ly | 3000 km | 1605 km | 1606 km |
| Lz | 23,100 m | 23,100 m | 23,100 m |
| DX | 10 km | 3 km | 2 km |
| DY | 10 km | 3 km | 2 km |
| CENTLAT | 43.0° N | 43.0° N | 43.0° N |
| CENTLON | 12.5° E | 12.5° E | 12.5° E |

A second grid is nested into the R10 grid. The nesting is one-way. For configuration C1, the second domain has 3 km horizontal resolution (R3), while for the second configuration (C2) the horizontal

resolution of the second grid is 2 km (R2). The two nested grids (R3, R2) have nearly the same domain extension in the NS and WE directions, which is about 1600 km. The vertical extension is the same for the R10, R3, and R2. The vertical grid counts 42 vertical levels, starting from the first level at 50 m above the surface in the terrain following coordinates used by RAMS@ISAC to a height of 23.1 km at the model top.

Microphysical parameterization scheme is WSM6. This scheme considers six hydrometeors, i.e., cloud, rain, graupel, snow, ice, water vapor. The exchange between surface, biosphere, and atmosphere is parameterized following the LEAF3 model [45]. This model includes prognostic equations for soil temperature and moisture for multiple layers, vegetation temperature and surface water, and temperature and water vapor mixing ratio of canopy air. The turbulent mixing in the horizontal directions is parameterized according to Smagorinsky [46], while vertical diffusion follows Mellor and Yamada [47]. Longwave and shortwave radiation is parameterized as in Chen and Cotton [48]. Cumulus convection parameterization follows the modified Kuo scheme, which accounts for updrafts and downdrafts (see Molinari and Corsetti [49]). This scheme is applied to the R10 grid only.

For the application of the cumulus parameterization scheme used in this paper, we assume that convective processes are explicitly resolved by the 3 km horizontal resolution. This assumption is widely accepted [50,51]. However, several studies investigated the impact of cumulus parameterization scheme at moderate to high horizontal resolutions (1–5 km) without reaching a general conclusion [52–54]. The latter point is confirmed by unpublished works with RAMS@ISAC model, which show that using cumulus parameterization at 3 km horizontal resolution can sometimes improve the forecast of convective systems. In this paper, the impact of lightning data assimilation when using the cumulus parameterization scheme is not quantified.

We prefer to avoid the activation of the cumulus parameterization scheme because a rapid increase of operational forecast horizontal resolution is foreseen in the future. For this reason, in this paper we considered also the 2 km horizontal resolution. As the horizontal resolution increases, the application of the cumulus parameterization scheme is less justified.

### 3.2. Lightning Data Assimilation

Lightning data are provided by LINET (LIghtning detection NETwork, [55]). LINET is a ground network with more than 500 detectors spread worldwide. More than 200 sensors are located in Europe, with a good coverage in Central Europe, including Italian territory and western Mediterranean Sea. LINET stations detect Very Low Frequency (VLF) and Low Frequency (LF) waves emitted during the flash, which permit to measure both intra-cloud (IC) and cloud-to-ground (CG) lightning. Signal timing is given through a GPS clock. Lightning detection method is based on a Pseudo-3D algorithm (TOA-3D; [56]), which consists of three phases: 2D location of the flash through a time of arrival algorithm (TOA), exploitation of the time delay at the sensor nearest to the lightning, and time relaxation of the travel path of the radio-wave. The position accuracy is of 125 m for an average distance of 200 km among sensors [55].

Lightning data assimilation in RAMS@ISAC model is performed following the scheme of Fierro et al. [23,24].

The practical implementation of the method works as follows: flashes are binned in 10 min time intervals starting from the initial simulation time. For each bin, all flashes inside a bin are assumed to occur at the starting time of the bin, ignoring the time differences among flashes. Every 10 min, i.e., for each bin, it is checked if there are flashes inside the grid cell centered at each grid point. If there are no flashes no action is made. If there are flashes, a pseudo-profile of water vapor mixing ratio is computed and nudged in RAMS@ISAC. The nudging relaxation time is 5 min. The pseudo-profile is a saturated mixing ratio profile from the lifting condensation level (LCL), determined from the model temperature and humidity fields, to the −25 °C level, also determined from model temperature.

The pseudo-profile has no data outside the layer between the LCL and −25 °C levels. This check is done every 10 min.

The method is like that implemented in Federico et al. [33], however there are two differences following recent developments of lightning data assimilation [6,57] and results of unpublished work with RAMS@ISAC.

First, in Federico et al. [33] the pseudo-profile of water vapor mixing ratio was determined as a function of the flash rate and graupel mixing ratio, following the works of Fierro et al. [23,24]. In this paper we consider a saturated mixing ratio profile. In Federico et al. [33], the nudging was performed for the vertical levels in the charging zone (0 °C/−25 °C), while in this work the nudging is performed between the lifting condensation level and -25 °C. Unpublished works with RAMS@ISAC show that the assimilation of water vapor starting from 0 °C level has often a lower performance compared to that starting from LCL, because the former underestimates intense precipitation.

*3.3. Numerical Experiments*

The VSF implementation is represented in Figure 6. First, a 36 h run of the model first domain (10 km horizontal resolution) starting at 12:00 UTC on 09 October 2018 was done. This simulation does not assimilate lightning and it is used to give initial and boundary conditions to the R3 and R2 grids to compensate for the large difference between large scale fields provided by general circulation models and R3 and R2 horizontal grid resolutions. The R10 simulation uses as initial and boundary conditions the 0.25° horizontal resolution Global Forecast System (GFS) analysis–forecast cycle issued at 12:00 UTC on 9 October 2018.

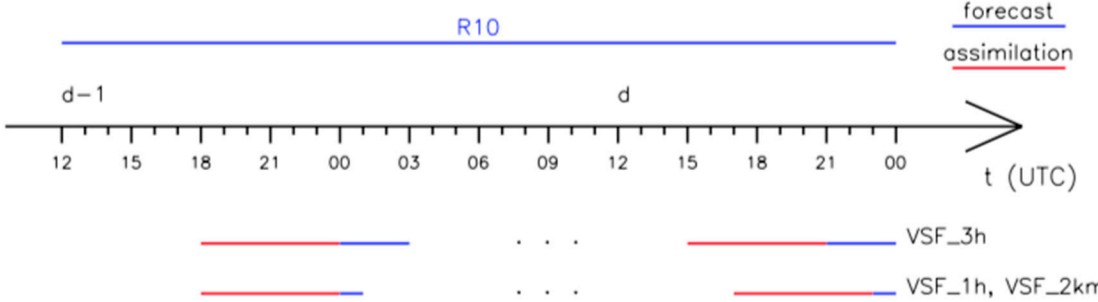

**Figure 6.** Scheme used for the Very Short-term Forecast (VSF) runs. For each line, the dots indicate that simulations between the first and the last (6 for VSF_3h; 22 for VSF_1h and VSF_2km) follow the same scheme of the two shown examples. The red color represents the assimilation phase while the blue one represents the forecast phase.

Then VSF runs are provided using the R10 simulation as initial and boundary conditions.

In this paper we considered four different numerical experiments at fine horizontal resolution (for experiments details see Table 2). All simulations, except the control simulation (CTRL), assimilate lightning. For the VSF_3h case, each simulation lasts 9 h; the first 6 h of each run are the assimilation period (also assimilation phase) and the last 3 h represent the forecast time (forecast phase), without assimilation. During the assimilation phase, lightning is assimilated every 10 min following the practical implementation outlined in the previous section. Eight VSF_3h forecasts are needed to span 10 October 2018.

For the VSF_1h and the VSF_2km cases, assimilation was performed as follows. Each simulation lasts 7 h; during the first 6 h flashes are assimilated, and then followed by a 1 h forecast. For these configurations 24 runs are necessary to cover 10 October 2018.

It is important to note that, in addition to initial and boundary conditions, lightning is the only observation assimilated by VSF_3h, VSF_1h, and VSF_2km. Of course, lightning could be assimilated in conjunction with other observations, which also improve the convection simulation. The contribution

of other observations could reduce the impact of lightning data assimilation in a data assimilation setting considering more data sources [21].

**Table 2.** Numerical experiments at fine horizontal resolution. $q_v$ is the water vapor mixing ratio. The assimilation phase is 6 h long for all numerical experiments.

| Numerical Experiment | Horizontal Resolution | Simulation Duration | Forecast Phase | Lightning Data Assimilation | Parameter Adjusted |
|---|---|---|---|---|---|
| CTRL | 3 km | 9 h | 3 h | No | None |
| VSF_3h | 3 km | 9 h | 3 h | Yes | $q_v$ |
| VSF_1h | 3 km | 7 h | 1 h | Yes | $q_v$ |
| VSF_2km | 2 km | 7 h | 1 h | Yes | $q_v$ |

## 4. Results

### 4.1. Phase 03–06 UTC on 10 October 2018

The precipitation of the phase 03–06 UTC was very intense in the eastern part of Sardinia where 142 mm/3 h were recorded in Tertenia and 67 mm/3 h were recorded in Flumini Uri a San Vito. More than 47 mm/3 h were observed in the central part of southern Sardinia (Figure 3a).

The large precipitation of this phase was not simulated by CTRL (Figure 7a), which missed the forecast. The VSF_3h (Figure 7b) did a much better forecast, simulating more than 60 mm/3 h over eastern Sardinia and 25 mm/3 h in the southern part of the Island.

While VSF_3h improves substantially the forecast of the intense precipitation of this phase when compared to CTRL, the precipitation amount is underestimated.

VSF_1h gives a considerable improvement to the rainfall forecast of VSF_3h (Figure 7c). The maximum rainfall over eastern Sardinia is 130 mm/3 h and the maximum over southern Sardinia is 70 mm/3 h. In both cases the precipitation amount is in good agreement with observations, showing two areas of intense precipitation over the island.

The increase of model resolution (Figure 7d) has also a significant impact on the precipitation forecast. The precipitation is reduced, compared to VSF_1h, both in eastern and southern Sardinia (100 and 50 mm/3 h, respectively). The reason for the precipitation reduction of VSF_2km compared to VSF_1h will be discussed more in detail in the next section.

The precipitation amounts simulated by VSF_1h and VSF_2km are in good agreement with observations while VSF_3h underestimates the rainfall prediction. While this result is expected because the 1 h forecast assimilates more frequently the flashes and follows the time evolution of the storm better, the important difference between the rainfall prediction of VSF_3h and that of VSF_1h (or VSF_2km) is, at least in part, caused by the inability of the RAMS@ISAC model to simulate and sustain the convection for this specific time period of the storm evolution. This is shown by (a) the very low precipitation simulated by CTRL; (b) the fact that once convection is forced by lightning data assimilation, like in VSF_3h, it decays rapidly, as shown by the considerably smaller amount of precipitation forecast by VSF_3h compared to VSF_1h and VSF_2km.

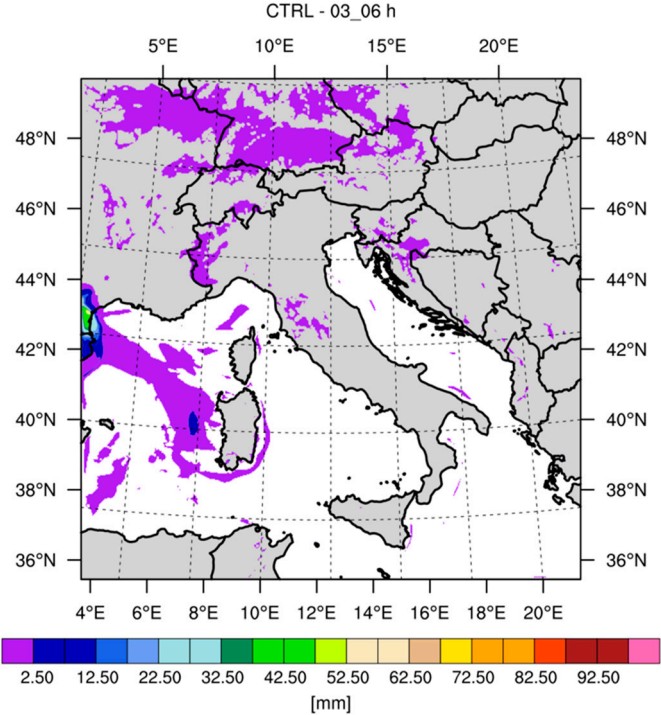

(a)

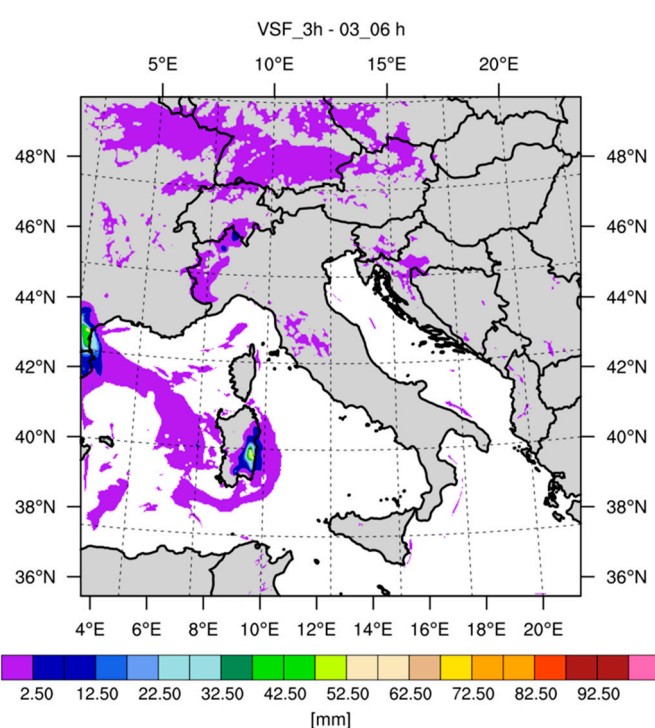

(b)

**Figure 7.** *Cont.*

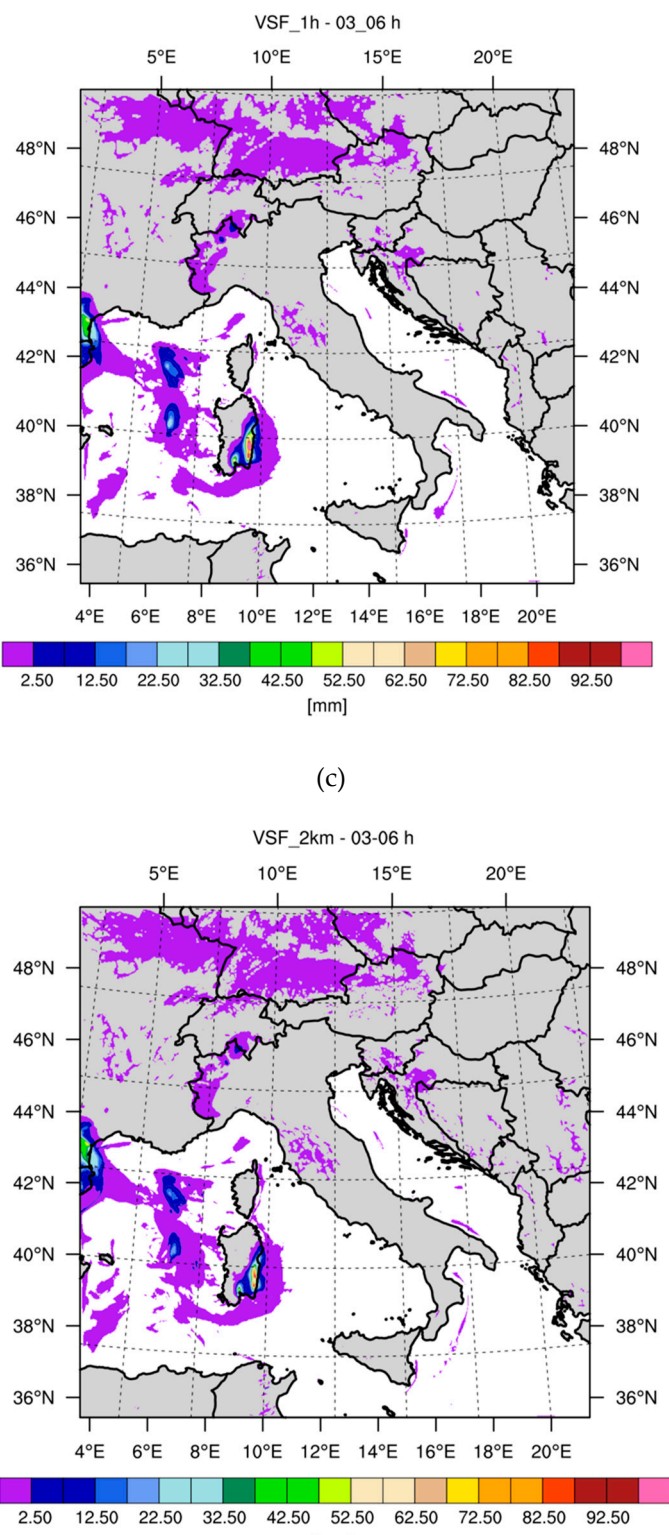

(c)

(d)

**Figure 7.** Accumulated precipitation between 03 UTC and 06 UTC on 10 October 2018 for (**a**) CTRL; (**b**) VSF_3h; (**c**) VSF_1h; (**d**) VSF_2km. For VSF_1h and VSF_2km the precipitation is given by the sum of the three very short term forecast of 1 h between 03 and 06 UTC.

To explore further the impact of lightning data assimilation on the VSF of this storm, the radar reflectivity simulated at 04 UTC and at 5 km height is considered in Figure 8 for the different model configurations. The 04 UTC time represents the end of an assimilation cycle for VSF_1h and VSF_2km, while it is 1 h inside the forecast phase for VSF_3h. The corresponding observation at 04 UTC is given in Figure 3b and shows values up to 50 dBz for the central part of southern Sardinia.

To compute the reflectivity from model output, we used the forward operator used in the RIP (Read/Interpolate/Plot) software of WRF (Version 3.9 [58]; see also the supplemental material of Federico et al. [20] for a description of the operator).

The reflectivity simulated by CTRL (Figure 8a) simulates a cloud over southern Sardinia, while it misses the reflectivity observed over eastern Sardinia. The VSF_3h represents the storm over eastern Sardinia while the reflectivity over southern Sardinia is underestimated. The comparison between observations and VSF_3h reflectivity shows that the model is not able to simulate properly the convection 1 h after the end of the assimilation stage because of the small value of reflectivity simulated over southern Sardinia. Additionally, the shape of the reflectivity is broader compared to the observations over eastern Sardinia.

The shape of the observed reflectivity over southern and eastern Sardinia is better represented by VSF_1h but the values of reflectivity are overestimated. The VSF_2km improves the reflectivity forecast compared to VSF_1h because the maximum reflectivity is lower and because the shape of the cloud system is narrower in VSF_2km, in better agreement with observations.

The simulation of the cloud system west of Sardinia island, over the sea, is not improved by lightning data assimilation and the simulated reflectivity is shifted to the northeast compared to the observations.

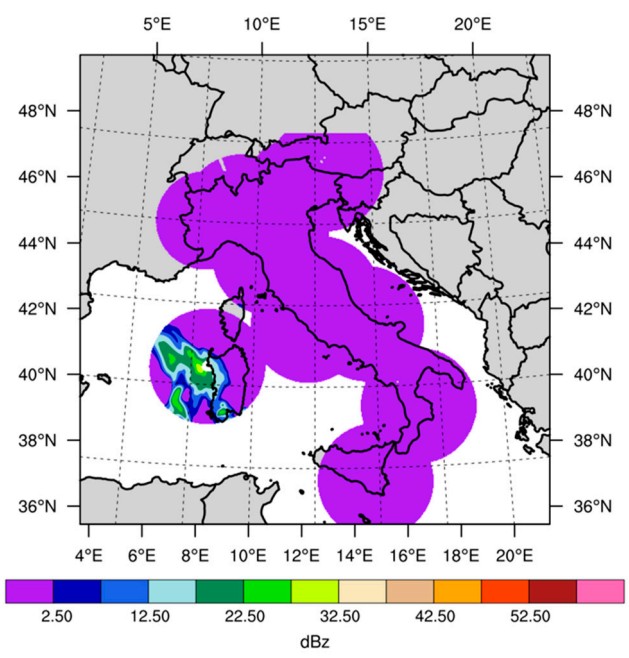

(a)

**Figure 8.** *Cont.*

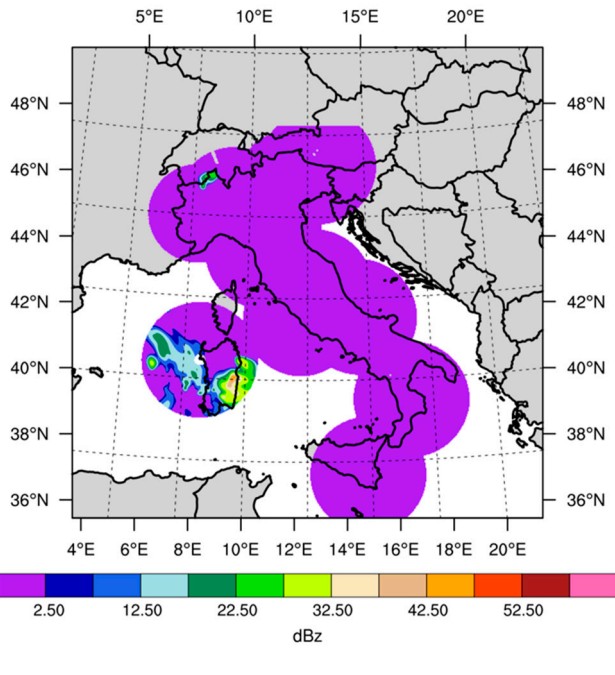

(b)

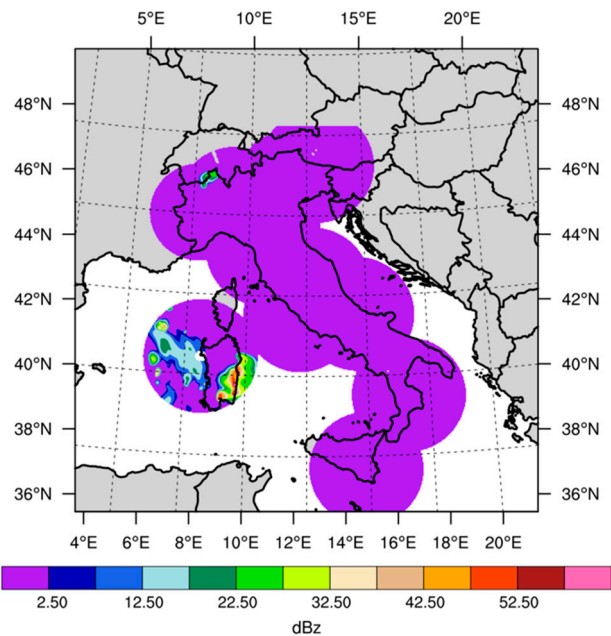

(c)

**Figure 8.** *Cont.*

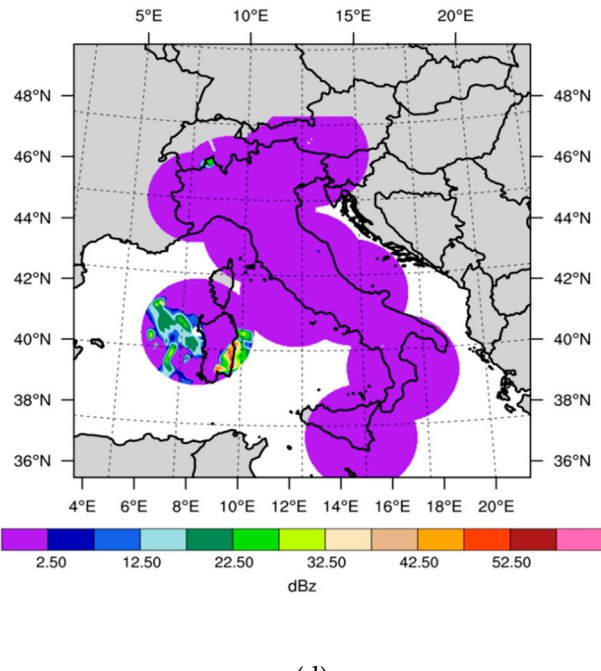

(d)

**Figure 8.** CAPPI simulated at 04 UTC and at 5 km by (**a**) CTRL; (**b**) VSF_3h; (**c**) VSF_1h; (**d**) VSF_2km.

The above discussion is confirmed by the scores for the simulated reflectivity computed at 04 UTC for three different reflectivity values (0, 5, 15 dBz) and for three heights (2, 3, 5 km, see Table 3; CAPPI are available at these heights for this case study). The poor performance of CTRL is shown by the low FBIAS (frequency bias), ETS (equitable threat score), and POD (probability of detection) for most heights and thresholds. At 5 km height, the FBIAS of CTRL is of the order of 1, nevertheless the POD remains low (<10%), showing that most forecasts are false alarms. The considerable impact of lightning data assimilation is apparent from the improvement of the ETS score for all thresholds and heights. The ETS score is improved by both the POD increase and the decrease of false alarms compared to CTRL. The performance of VSF_1h is better than VSF_3h.

It is important to note that the FBIAS of simulations assimilating lightning is overestimated at 3 and 5 km heights. This tendency is partially reduced by VSF_2km.

*4.2. Phase 21 UTC on 10 October 2018–00 UTC on 11 October 2018*

In this phase, the precipitation was recorded in the eastern and southern parts of Sardinia. The rainfall intensified over southeastern Sardinia where a maximum of 76 mm/3 h was recorded in Villasalto. Rainfall was recorded also over Liguria and Piedmont regions. This precipitation was caused by the forerunner of a meteorological system that brought severe weather over northern and central Italy on the following day (11 October).

Differently from the phase analyzed in the previous section, the CTRL run (Figure 9a) performs well. Maximum precipitation forecast is more than 60 mm/3 h showing the severity of the storm over southeastern Sardinia during this phase. The rainfall forecast of VSF_3h shows higher amounts compared to CTRL, as expected because the implementation of lightning data assimilation adds water vapor to the forecast during the assimilation phase. The maximum observed precipitation over Sardinia is between that of VSF_3h (100 mm/3 h) and that of CTRL (60 mm/3 h). Moreover, the location of maximum rainfall is between those simulated by CTRL and VSF_3h. The 1 h very short-term forecast (VSF_1h) increases the precipitation over southeastern Sardinia giving a considerable amount of false alarms. Nevertheless, the VSF_1h represents better the rainfall over Liguria. Increasing the

horizontal resolution (VSF_2km) has a positive impact on the rainfall forecast mainly because the rainfall overestimation is reduced compared to VSF_1h.

**Table 3.** Scores of simulated reflectivity for three different reflectivity thresholds (0, 5, and 15 dBz) and for three observation levels (2000, 3000, and 5000 m). Scores are computed for the four VSF configurations considered in the paper (CTRL, VSF_3h, VFS_1 h, VSF_2km). FBIAS is the frequency bias, ETS is the equitable threat score, POD is the probability of detection, and FAR is the false alarm rate.

| dBz_th = 0 | | | | | | | | | | | | |
|---|---|---|---|---|---|---|---|---|---|---|---|
| | FBIAS | | | ETS | | | POD | | | FAR | | |
| Height (m) | 2000 | 3000 | 5000 | 2000 | 3000 | 5000 | 2000 | 3000 | 5000 | 2000 | 3000 | 5000 |
| CTRL | 0.02 | 0.30 | 1.45 | 0.0 | 0.07 | 0.10 | 0.0 | 0.12 | 0.43 | 1.0 | 0.60 | 0.70 |
| VSF_3h | 1.22 | 1.52 | 2.20 | 0.17 | 0.20 | 0.28 | 0.42 | 0.55 | 0.92 | 0.66 | 0.64 | 0.58 |
| VSF_1h | 1.09 | 1.34 | 1.85 | 0.18 | 0.21 | 0.33 | 0.40 | 0.52 | 0.88 | 0.63 | 0.61 | 0.52 |
| VSF_2km | 0.81 | 1.01 | 1.45 | 0.15 | 0.20 | 0.35 | 0.30 | 0.42 | 0.76 | 0.62 | 0.58 | 0.47 |
| dBz_th = 5 | | | | | | | | | | | | |
| | FBIAS | | | ETS | | | POD | | | FAR | | |
| Height (m) | 2000 | 3000 | 5000 | 2000 | 3000 | 5000 | 2000 | 3000 | 5000 | 2000 | 3000 | 5000 |
| CTRL | 0.01 | 0.07 | 1.39 | 0.0 | 0.0 | 0.05 | 0.0 | 0.0 | 0.31 | 1.0 | 0.95 | 0.78 |
| VSF_3h | 1.20 | 1.54 | 2.40 | 0.17 | 0.18 | 0.27 | 0.40 | 0.51 | 0.93 | 0.67 | 0.67 | 0.61 |
| VSF_1h | 1.09 | 1.38 | 1.90 | 0.19 | 0.21 | 0.35 | 0.40 | 0.51 | 0.89 | 0.63 | 0.63 | 0.53 |
| VSF_2km | 0.84 | 1.12 | 1.60 | 0.16 | 0.21 | 0.36 | 0.31 | 0.45 | 0.80 | 0.63 | 0.60 | 0.50 |
| dBz_th = 15 | | | | | | | | | | | | |
| | FBIAS | | | ETS | | | POD | | | FAR | | |
| Height (m) | 2000 | 3000 | 5000 | 2000 | 3000 | 5000 | 2000 | 3000 | 5000 | 2000 | 3000 | 5000 |
| CTRL | 0.01 | 0.02 | 1.05 | 0.0 | 0.0 | 0.02 | 0.0 | 0.0 | 0.13 | 1.0 | 1.0 | 0.88 |
| VSF_3h | 2.23 | 2.69 | 3.05 | 0.09 | 0.11 | 0.19 | 0.35 | 0.47 | 0.80 | 0.84 | 0.82 | 0.74 |
| VSF_1h | 2.15 | 2.37 | 2.78 | 0.15 | 0.16 | 0.24 | 0.48 | 0.56 | 0.87 | 0.78 | 0.76 | 0.69 |
| VSF_2km | 1.69 | 1.92 | 2.33 | 0.13 | 0.16 | 0.27 | 0.37 | 0.46 | 0.82 | 0.78 | 0.76 | 0.65 |

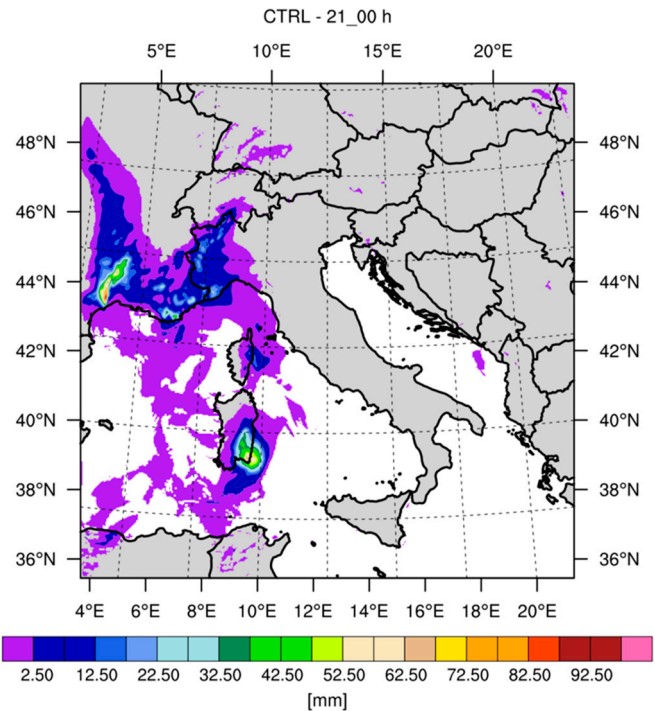

(a)

**Figure 9.** *Cont.*

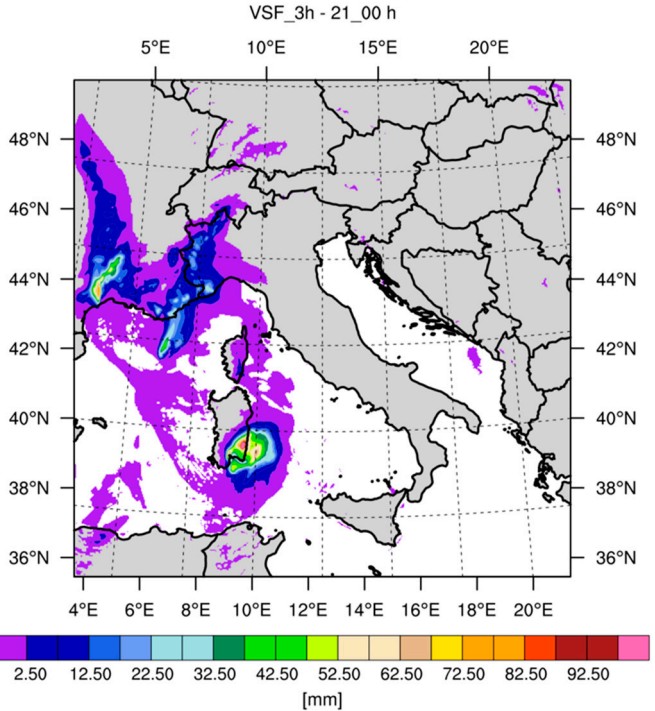

(b)

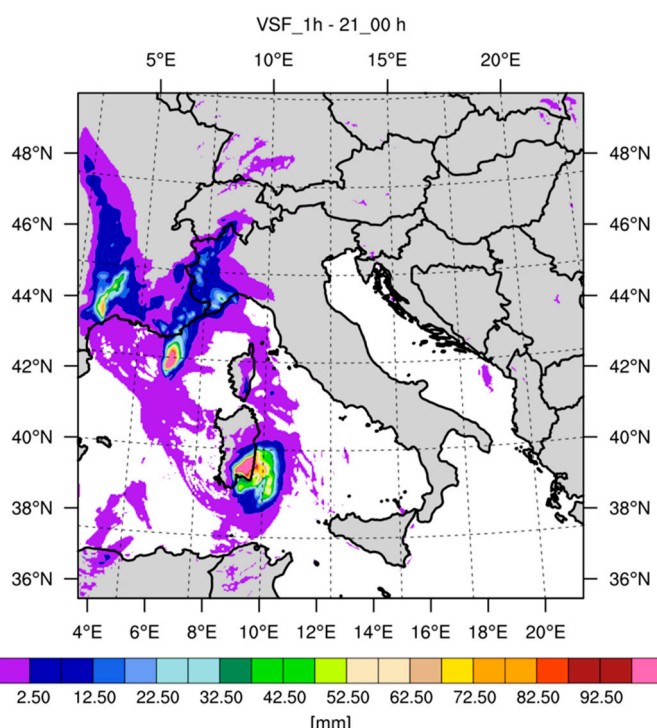

(c)

**Figure 9.** *Cont.*

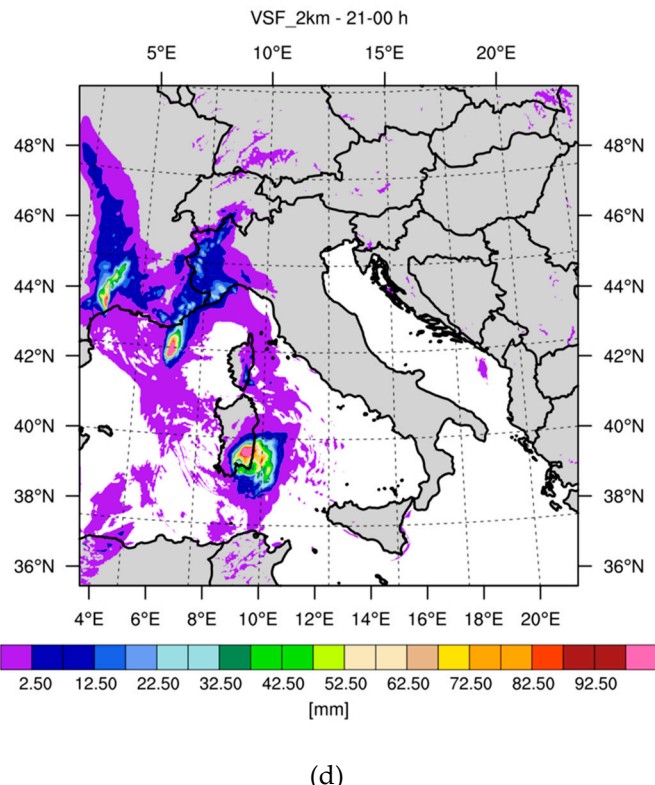

(d)

**Figure 9.** Accumulated precipitation between 21 UTC on 10 October 2018 and 00 UTC on 11 October 2018 for (**a**) CTRL; (**b**) VSF_3h; (**c**) VSF_1h; (**d**) VSF_2km. For VSF_1h and VSF_2km the precipitation is given by the sum of the three very short term forecast of 1 h between 21 and 00 UTC.

Changes in the horizontal resolution of simulations assimilating lightning produce a significant impact on the rainfall forecast, as shown by both phases of the Sardinia case considered in this and in the previous section. This is caused not only by the different representation of the atmospheric processes when horizontal resolution is changed but also by the differences in lightning data assimilation. In fact, the scheme for assimilating lightning considers, for each grid point, if flashes are recorded in a grid box centered at this grid point. If there are flashes, the assimilation is performed. Due to the different horizontal resolution, the number of grid points (and the areas) affected by lightning data assimilation changes between VSF_1h and VSF_2km.

The important impact of horizontal resolution on the water vapor is quantified in Figure 10, which shows the averaged water vapor between 3 and 10 km for VSF_1h (Figure 10a) and the difference of this quantity between VSF_1h and VSF_2km (Figure 10b) at 22 UTC on 10 October 2018, at the end of the assimilation cycle. Two points are worth noting: (a) there is a pattern shift between the two simulations and VSF_1h has a larger water vapor amount over southern Sardinia; (b) the amount of water vapor for VSF_1h is larger than that of VSF_2km (i.e., net value of the field of Figure 10b is positive when integrated over the domain shown in the figure).

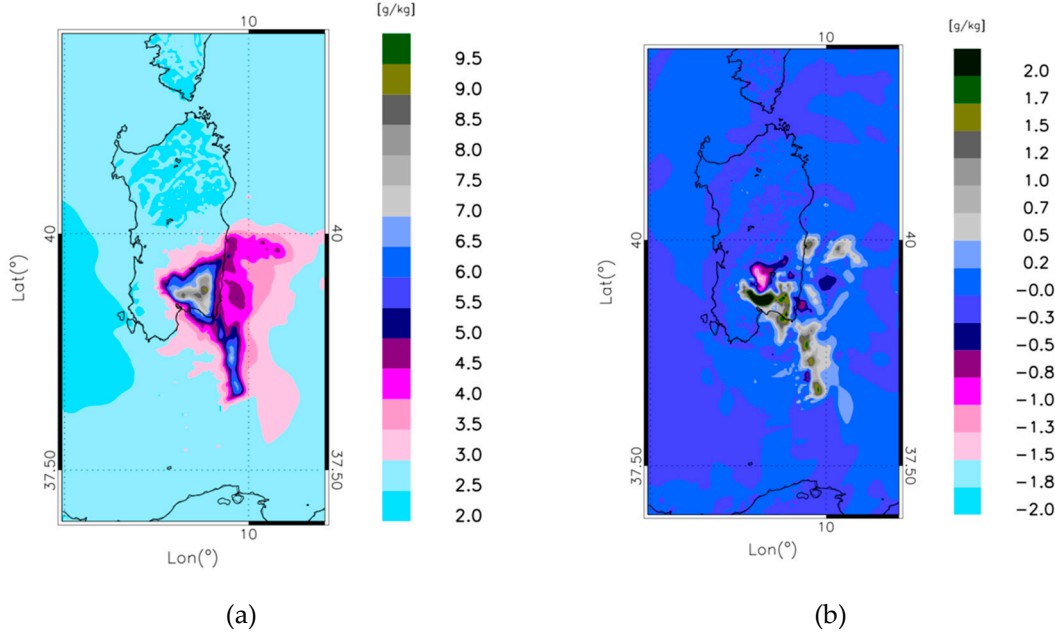

(a)　　　　　　　　　　　　　　　　　　　　　(b)

**Figure 10.** (**a**) Averaged water vapor between 3 and 10 km for VSF_1h (a zoom over Sardinia is provided for clarity) at 22 UTC on 10 October 2018; (**b**) difference between the averaged water vapor of VSF_1h and VSF_2km at the same date and time. The averaged water vapor between 3 and 10 km of VSF_2km was interpolated, using bilinear interpolation, onto the 3 km grid used by VSF_1h.

## 5. Performance Diagram for 10 October 2018

In this section we consider the performance diagram computed for the rain gauges over Sardinia island (Figure 11a) and for rain gauges over the whole of Italy (Figure 11b) for 10 October 2018. The verification over Sardinia is presented to focus on the impact of lightning data assimilation over the island, while the performance over Italy is presented to account for the precipitation that occurred over northwestern Italy. The verification was done for precipitation accumulated in 3 h time intervals considering all the pairs (observation; forecast) for all the 3 h forecast of the day (00–03 UTC, 03–06 UTC, . . . 21–00 UTC). For the simulations VSF_2km and VSF_1h, the 1 h forecasts are summed in 3 h forecasts for each period. For example, if we consider the phase 00–03 UTC, the 3 h precipitation forecast of VSF_1h (or VSF_2km) is obtained by summing the precipitation forecast of 00–01 UTC, 01–02 UTC, and 02–03 UTC.

The verification was done considering the nearest neighborhood method. In this method, for each rain gauge, we searched for the precipitation forecast closest to the observed value inside a fixed search radius. In Figure 11 the search radii are $2\Delta x \sqrt{2} \simeq 8.5$ km, where $\Delta x$ is the grid spacing of R3, i.e., 3 and 20 km.

The performance diagram for Sardinia shows the notable impact of lightning data assimilation on the rainfall forecast. In fact, CTRL has the worst score for the thresholds and nearest neighborhood radii considered. It is also notable the difference between the 3 h forecast (VSF_3h) and 1 h forecast (both VSF_1h and VSF_2km).

The comparison between VSF_1h and VSF_2km shows that the former has larger POD and FAR (false alarm rate). This is caused by the more abundant precipitation forecast produced by VSF_1h with respect to VSF_2km. Notably, the performance of VSF_1h and VSF_2km is high because the POD is larger than 0.8 and the FAR is less than 0.5. This shows that lightning data assimilation can locate convection and precipitation precisely in space and time.

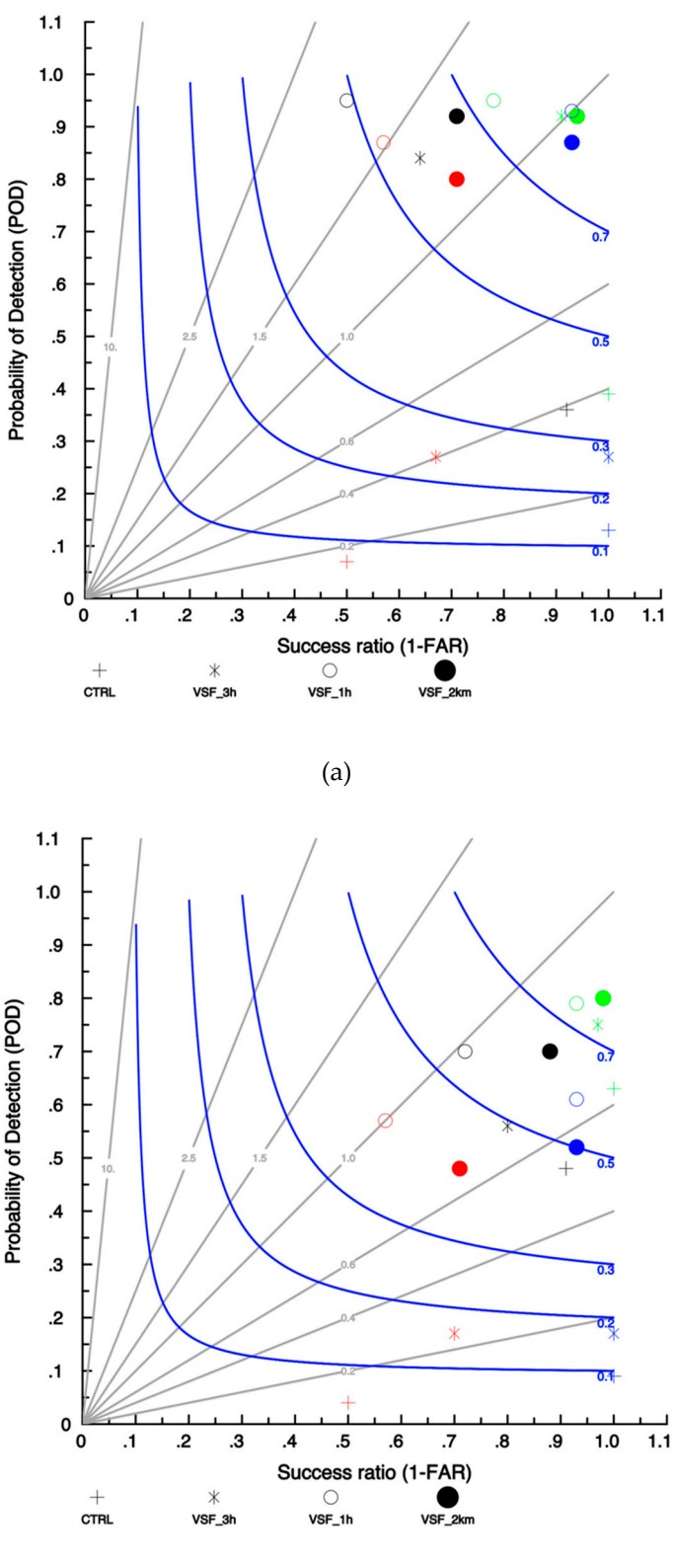

(a)

(b)

**Figure 11.** Performance diagram for the Sardinia rain gauges (**a**) and for the whole of Italy (**b**). Black markers are for the 10 mm/3 h threshold and for 8.5 km nearest neighborhood, red markers are for the 50 mm/3 h threshold and for 8.5 km nearest neighborhood, green markers are for the 10mm/3 h threshold of the 20 km nearest neighborhood, and blue markers are for the 50 mm/3 h threshold and 20 km nearest neighborhood.

Considering the results over Italy, a decrease of the performance is noticed for the simulations assimilating lightning, while the CTRL improves its performance for the 10 mm/3 h threshold thanks to a good prediction of the precipitation over the northwest of Italy for this threshold. In general, however, the difference between the CTRL forecast and the simulations assimilating lightning is apparent. Most of the time, the simulation VSF_2km gives the best forecast.

In the numerical experiments of this paper the spin-up time is 6 h, which could pose an issue because a moisture-related variable is assimilated. However, the results of the numerical simulations show that the performance of 1 h VSF is better than that of 3 h VSF. This shows, indirectly, that 6 h spin-up is enough for the case study considered because, if the spin-up time had been an issue, the 3 h VSF would have been better than the corresponding 1 h VSF. Similarly, Federico et al. [34], using 6 h spin-up time, showed that the 3 h VSF was better than the 6 h VSF which, in turn, was better than the 12 h VSF for 20 case studies over Italy. The systematic worsening of the precipitation forecast as the forecast time increases confirms that 6 h spin-up time is enough for the model to equilibrate imbalances between dynamic and thermodynamic fields.

## 6. Conclusions

In this paper we showed the performance of RAMS@ISAC model for a challenging case study occurred over Sardinia island on 10 October 2018. The forecast of the rainfall of the event was challenging because precipitation was intense and localized in two spots over eastern and south-central Sardinia.

The CTRL forecast, without lightning data assimilation, gives a good forecast only for the last very short-term forecast of the event (21 UTC on 10 October–00 UTC on 11 October), while observations show high rainfall intensity for most of the day.

The lightning data assimilation has a significant impact on the model performance. The VSF_3h, which assimilates flashes for 6 h and then performs a 3 h forecast, simulates the deep convection and rainfall occurring over eastern and south-central Sardinia for the whole day. Nevertheless, for some phases, the VSF_3h forecast is not completely satisfactory because intense precipitation spells are missed.

To improve the forecast, the sensitivity to a more frequent assimilation cycle was studied. In particular, the 1 h VSF (both VSF_1h and VSF_2km, which does a 1 h forecast after assimilating lighting for 6 h) improved substantially the precipitation forecast compared to 3 h VSF.

The significant difference between VSF_3h and simulations performing 1 h forecast, both VSF_1h and VSF_2km, is caused by the sudden decay of convection in VSF_3h simulation for some phases of the storm. Convection is restored by lightning data assimilation in VSF_1h and VSF_2km, improving the precipitation forecast.

However, when the CTRL run correctly simulates the convection, as between the 21 UTC on 10 October and 00 UTC on 11 October, the VSF_1h overestimates the rainfall.

Since this meteorological event was characterized by localized and heavy rainfall, the horizontal resolution is an important aspect of the forecast. For this reason, we considered the impact of the horizontal resolution on the precipitation forecast with simulation VSF_2km, which has a 2 km horizontal resolution. As expected, the impact of increasing the horizontal resolution on the rainfall forecast is considerable and is also caused by the differences of lightning data assimilation between VSF_2km and VSF_1h. For this case study, the simulation with 2 km horizontal resolution has a lower water vapor amount at the end of the assimilation phases than VSF_1h. This causes less precipitation and a decrease of false alarms for VSF_2km compared to VSF_1h, improving the forecast of VSF_2km.

The problem of false alarms was highlighted in several papers considering the forecast of severe weather and lightning data assimilation. This study shows the important role of the model horizontal resolution in reducing false alarms. Nevertheless, to quantify this impact robustly, a much larger number of cases must be considered.

**Author Contributions:** Conceptualization, R.C.T., S.F., and S.D.; methodology, R.C.T., S.F. and S.D.; software, R.C.T. and S.F..; validation, R.C.T., S.F. and A.C.P.; formal analysis, R.C.T., S.F., A.C.P.; data curation, S.P., G.V.; writing—original draft preparation, R.C.T. and S.F.; writing—review and editing, all authors; funding acquisition, S.D., S.F., S.P. and G.V. All authors have read and agreed to the published version of the manuscript.

**Funding:** This research was partially funded by the agreement between the Italian Department of Civil Protection and ISAC-CNR.

**Acknowledgments:** Part of the computational time used for this paper was granted by the ECMWF (European Centre for Medium range Weather Forecast) throughout the special project SPITFEDE. LINET data were provided by Nowcast GmbH (https://www.nowcast.de/) within a scientific agreement between H.D. Betz and the Satellite Meteorological Group of CNR-ISAC in Rome.

**Conflicts of Interest:** The authors declare no conflict of interest.

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
