# Peer review of "Application of Lightning Data Assimilation for the 10 October 2018 Case Study over Sardinia"

_atmosphere, doi:10.3390/atmos11050541_

Round 1

Reviewer 1 Report

In the manuscript the predictive power is assessed of severe-weather forecast and considerable improvements are shown by assimilating lightning data.

This is a topic of much current interest and several different groups have shown that assimilating lightning data gives a considerable improvement of the short term weather prediction. The present work follows rather closely that of reference 19. The details of the used method for assimilating lightning data are different, but from the publication it is not clear whether or not these differences give a significant improvement, nor if the present method outperforms other methods that have been published in recent years.

The over view of the existing literature on the topic is very superficial and is missing many recent works.

My recommendation is therefor to improve on the review of existing works and state very clearly what the added value is of the present investigation, what is distinguishing the present assimilation method from others in the literature (not only ref 19), why a different approach is taken, and how the present approach outperforms the others.

Author Response

Thank you for the helpful revision. Find attached our response.

Reviewer 2 Report

This ms examines the applicability of lightning data assimilation to a case study over Sardinia. It is found that lightning DA yields improvements over the control forecast.

I have some minor concerns (described below) but I think the ms can be accepted after suitable revisions.

l.53 "high-resolution numerical weather prediction models (< 3-5 km horizontal resolution), which explicitly resolve convective precipitation, the latter resolution being the focus of this paper" My understanding is that some operational models are still run with a convective parameterisation up to ~1 km, though of course details may vary from model to model.

l.80 precipitation forecast of a challenging case occurred-> precipitation forecast of a challenging case that occurred

l.81 precipitation spots -> precipitation events?

l.139 didn't -> did not. Contractions are used in several other places.

l.200 "Cumulus convection parameterization follows the modified Kuo scheme, which 201 accounts for updrafts and downdrafts (see Molinari and Corsetti [33]). This scheme is applied to the R10 grid only." As discussed above, 3 km horizontal resolution may be too coarse for a convection-permitting resolution. This raises the question, therefore, as to whether the impact of lightning DA would be as great when a convective scheme is present. Please discuss. 

l. 243 ff. (i) I think it would be helpful to list the different runs (including CTRL) in a table. Since explicit descriptions are not given, a certain amount of guesswork is required to determine how the runs differ from each other. (ii) Is any DA performed for the CTRL forecast? Presumably not. The text simply mentions that the runs are initialised from the R10 simulation. If so, the effect of the lightning DA could be overstated. In practice, the lightning DA would likely be performed along with assimilation of other variables. To put matters differently, "regular DA" could yield comparable improvements to lightning DA. It would be helpful if the authors discussed this issue.

l.244 "the first 6 hours of each run are the assimilation period (also assimilation phase)" Does this mean that latent-heat nudging is performed continuously over this time? On l.223 it's stated that "this check is done every ten minutes." Since latent-heat nudging was originally derived for 3D-Var, the temporal aspect of the assimilation should be clarified.

Fig. 6 I don't entirely understand this figure. (i) Why are there gaps for VSF_3h and VSF_1h? (ii) What do the dotted lines stand for?  

Sec. 4 Regarding the interpretation of the results, is there any evidence of spin-up effects? This can be an issue when assimilating moisture-related variables.

Author Response

We thanks the reviewer for the very useful comments, which improved the quality of this paper. Our detailed answers are in the attacjed file.

Round 2

Reviewer 1 Report

The authors have made considerable improvements in the overview of the literature on the subject. I feel this will help to improve the impact of the paper. 

It is too bad that the authors have not been able to make a comparison between some of the many assimilation procedures.